# HIV UTR, LTR, and Epigenetic Immunity

**DOI:** 10.3390/v14051084

**Published:** 2022-05-18

**Authors:** Jielin Zhang, Clyde Crumpacker

**Affiliations:** Department of Medicine, Beth Israel Deaconess Medical Center, Harvard Medical School, Boston, MA 02215, USA

**Keywords:** HIV, super enhancer, epigenetics, chromatin vaccine (cVaccine), cure

## Abstract

The duel between humans and viruses is unending. In this review, we examine the HIV RNA in the form of un-translated terminal region (UTR), the viral DNA in the form of long terminal repeat (LTR), and the immunity of human DNA in a format of epigenetic regulation. We explore the ways in which the human immune responses to invading pathogenic viral nucleic acids can inhibit HIV infection, exemplified by a chromatin vaccine (cVaccine) to elicit the immunity of our genome—epigenetic immunity towards a cure.

## 1. Introduction

The onset of HIV/AIDS in the 1980s is a naturally occurred loss of function model in studying of the human immunity to find the prevention and cure. More than four decades of research has contributed multidisciplinary knowledge ranging from the viral infection to human immunity, including, but not limited to, the HIV lifecycle and the role of CD4 T-cells in the host immunity, as well as how HIV maneuvers the host transcriptional machinery, specially releasing a pause of RNA polymerase II (RNAPII) in transcription of the viral RNA [1,2,3].

The study of HIV/AIDS has taught us that from a molecular to a systemic level, human DNA has evolved a defense system against the infection of pathogenic nuclei acids. Herein, we review the contributions of many investigators, aiming to assemble the knowledge in a reminiscing and holistic manner [4,5,6,7,8,9], and to provide some foresight in production of effective immunogens such as the chromatin vaccines (cVaccines) for eliciting the host genetic immunity against HIV infection. 

Since structure determines function in biology, we first review the studies on the HIV UTRs, which are the un-translated regions of HIV RNA. Both HIV 5′ and 3′ UTR have regulatory roles in the AIDS pathogenesis. Next, we will follow with studies on the HIV LTRs, the DNA elements where the HIV enhancer and promoter are located. It is well recognized that the HIV 5′ and 3′ LTR play indispensable roles in HIV replication that is called the viral load, as well as in the latent infection that is dubbed the viral reservoir. Finally, we discuss these findings in a context of host epigenetic regulation, which embodies a host genetic immune system against the infection of HIV. We have termed this immunity of DNA genome as the epigenetic immunity [10]. 

Differing from innate, adaptive, and trained immunities, epigenetic immunity occurs in every eukaryotic cell that has a nucleus. Epigenetic immunity consists of three elements protecting our DNA genome: DNA methylation, histone modification, and ncRNA activity. Upon antigen stimulations, which are usually foreign nucleic acids, a host genome launches immune responses via epigenetic regulations consisting of DNA methylation, histone modification and noncoding RNA (ncRNA) function to protect the DNA integrity. 

## 2. HIV UTRs

HIV is a retrovirus, and its replication cycle consists of viral RNA and viral DNA. The viral DNA embedded in the human DNA is called the provirus. HIV RNA is a positive, genomic RNA (gRNA), and it encodes all the viral proteins but needs a reverse transcription to integrate into our genome, where it becomes a provirus, i.e., as a part of our DNA. 

Similar to the classic cellular mRNA, HIV RNA has both 5′UTR and 3′UTR. The differences, however, are the following: (1) both HIV 5′ and 3′UTR are longer than their cellular mRNA counterparts. (2) The average length of human mRNA is 3.4 kb. HIV-1 mRNAs lie in a range from 9.3 kb of gRNA to the 2 kb of spliced mRNA, which encode the HIV structural and accessory proteins. All the HIV mRNAs possess a similar 5′ and 3′ UTR region.

### 2.1. 5′UTR

The 5′UTR of HIV-1 gRNA is known to form specific structures and has important functions besides those of classic cellular mRNA [4]. These include TAR, the elongating sequences required for gRNA replication [11,12,13,14], site of gRNA dimerization (in kissing or linear form) [15], packaging signal (ψ) [16,17,18], site of reverse transcription initiation (tRNA_Lys3_ annealing) [19], and a cap- or an internal ribosome entry site (IRES) for cap-independent translational initiation [20,21,22,23]. 

Mutations in the sequence of 5’UTR disrupt the structure of this region and affect the transcription of HIV RNA, reverse transcription, and packaging, as well as the fundamental role of mRNA in protein translation, i.e., the viral protein production. Note that starting at transcription, more than 23 copies of HIV mRNA are produced, which include alternatively spliced isoforms of env mRNA for translation of the regulatory, accessory, and Env proteins, and the gRNA for Gag-Pol structural polyproteins [24,25,26,27]. 

The multiple splice donor (SD) and splice acceptor (SA) sites in the HIV-1 gRNA support alternative splicing for a pool of 4-kb and 2-kb mRNAs that differ in their 5′UTR. These mRNAs contain partly overlapping open reading frames (ORFs), and collectively encode the nine HIV-1 proteins and polyproteins [24,25,26,27]. In addition, the IRES-dependent translational initiation is a cap-independent translational process. An internal AUG codon found near the amino terminus of the Pr55(gag) capsid domain drives translation of a 40-kDa Gag isoform [20,21,22,23,27]. 

The 5′UTR has been found to be the most conserved part in the HIV gRNA. In simple terms, its 335-nucleotide residues form the regulatory motifs, which initiate multiple steps in the HIV lifecycle. For example, the HIV-1 5’UTR reveals a 3D tRNA mimicry and is important for the viral reverse transcription. The 3D structure, but not the RNA sequence, is conserved across the distinct HIV-1 subtypes such as in C, B and A [28]. The tertiary motif is correlated to the region in 5′LTR after a reverse transcription for the formation subtypes of HIV [28,29,30]. 

### 2.2. 3′UTR

HIV-1 has a longer 3′UTR compared to the cellular mRNA. The HIV-1 encodes a polyadenylation (polyA) signal (AAUAAA) within both its highly conserved 5′ and 3’ UTR sites [31,32]. Note that in the polyadenylation process, an RNA transcript is cleaved and then elongated with the polyA. With repression of the 5’ polyA signal, the utilization of the 3’ polyA signal occurs. Studies suggest that polyA signals in the HIV RNA show metastable features to switch into different structures that regulate the viral gRNA function [31]. 

In addition to the classic role of 3′UTR in translational termination and the stability of RNA, HIV 3′UTR plays a cardinal role in formation of the HIV provirus. This is a separate topic, however, and will not be addressed in this review. Besides the role in HIV provirus formation, the 3′UTR takes part in the HIV packaging, a process of virion formation. 

Studies have shown that among pools of cellular mRNAs, those with a long 3’UTR, including the HIV RNA, are selectively packaged into the virion [33]. It seems plausible that the 3’UTR, a stretch of RNA not occupied by ribosomes, offers a favorable binding site for Gag, through encapsidation to the virion packaging. Moreover, Rev protein has been shown to function at the HIV-1 3′UTR. The Rev-responsive element (RRE) overcomes the inhibitory effects of a 5’ splice site located within the 3’UTR, to transport the HIV RNA from nucleoplasm to the cytoplasm [34]. 

Previous studies report that the host factors, e.g., eukaryotic translation initiation factor 3 subunit F (elF3f), restrict the HIV mRNA expression and target in the 3′UTR region, binding here blocks 3′ end processing [35,36]. Cellar miRNAs and HIV-encoded miRNAs, both mediating the repression of HIV RNA, are also targeting on the 3′UTR [37,38]. Recent discoveries on the epigenetic regulation of 3′UTR have further broadened our knowledge of host immunity in control of viral RNA expression via the 3′UTR [39,40]. 

Beside the role of miRNA in repression of the HIV RNA expression, the modification on viral 3’UTR m(6)A sites, analogous the cellular m(6)A sites, strongly enhances mRNA expression in cis by recruiting the cellular YTHDF m(6)A “reader” proteins. This protein-RNA interaction has shown a cell type specific manner. For example, studies show that over expression of YTHDF, originally found in *Drosophila* where m6A recruits YTHDF to the mRNA, enhances the HIV RNA, protein expression and the viral replication in the CD4 T-cells [39]. YTHDF2 has recently been shown to increase HIV mRNA stability while YTHDC1 reader recruitment helps regulate HIV alternative splicing [41].

Studies on viruses other than HIV have also shown the cell type specific expression of a virus. In other words, a virus replicates only in a specific type of cells—the viral or cell tropism. These studies unveil that the cell type specific viral replication is also driven by the viral mRNA 3′UTR, where the host cell factors interact with the viral 3′UTR motifs to stabilize viral protein expression [40,42]. 

Furthermore, ncRNA has been shown to act on the 3′UTR with the anti-viral effect. An artificial miR-30a-3’-untranslated region (miR-3-UTR) obtained from a single RNA polymerase II (RNAP II) was used to simultaneously target all HIV transcripts. The study reports that HIV-1 replication was significantly inhibited in the cells with the miR-3-UTR construct that acts as the RNA interference (RNAi) [43]. A highly abundant miRNA, miR-29a, is also reported to specifically target the HIV-1 3’UTR region, and enhances the viral mRNA association with RNAi machinery RISC (RNA-induced silencing complexes) and P body proteins to inhibit the HIV RNA expression [44]. 

To summarize, besides its important biological function in the viral lifecycle, the 3′UTR is a target of host antiviral immunity that functions at the epigenetic level. Of note, some of the ncRNA are transcribed directly from the viral 3′UTR, and target this region to induce HIV specific RNAi. The ncRNA, exemplified by miRNAs, bind via a protein complex to the HIV 3’UTR. RNAi is one of the epigenetic silencing mechanisms. 

The ncRNA function is one of the three elements of epigenetic immunity, and the 3′UTR, either of viral RNA or ncRNA, has a paramount role in the modulation of host immunity against the HIV infection. Note that for simplicity, the ncRNA includes miRNA, siRNA, shRNA, snRNA, lncRNA, and eRNA (enhancer RNA). 

## 3. HIV LTRs

After reverse transcription, the 9.3 kb gRNA becomes the HIV DNA to be integrated into the host DNA with an integration process. Of note, through reverse transcription, the 5′UTR becomes the 5′LTR, and 3′UTR becomes the 3′LTR. Upon a successful integration, the provirus is produced. 

Provirus not only keeps all the regulatory regions highly conserved in the 5′ and the 3′UTR, but also a new U3 region is in the front of the 5′LTR and a new U5 is at the end of the 3′LTR (Figure 1). This results in a longer DNA sequence than the 5′ and 3′UTR of gRNA and confers the long terminal repeat (LTR) to the provirus. A provirus contains the duplicated LTR sequences, meaning two identical enhancer-promoter regions, flanking the proviral genome with a 5’ and 3’ LTR symmetry. The LTR has one specific role—to transcribe the HIV RNA. 

It is easily understood that 5′ and 3′ LTR govern HIV RNA expression. The provirus is a parasite, and the transcription of gRNA relies on the host. In fact, provirus is a metastable stage in the HIV lifecycle, and the break of a metastable equilibrium depends on the host, not the parasite. In other words, a competition occurs between the host immunity and the virus on the viral gene expression—the transcription of the viral RNA. In this competition, HIV has 5′ and 3′ LTR, but the host has an entire genomic DNA, which has ~3.2 × 10^9^ nucleotide base pairs deploying DNA methylation, histone modification, and ncRNA function against the viral RNA replication.

After more than 40-years, HIV/AIDS research has led to an explosion of knowledge on human immunity, based on the fact that HIV/AIDS is a naturally occurring model of loss of function in the human immune response. The study of HIV/AIDS has provided an unprecedented platform to understand our immunity, specifically the role of CD4 T-cells in eukaryotic vs. prokaryotic immunity against viral infections, and in prevention as well as cure, including the vaccination (immunization). 

In essence, research on HIV/AIDS has unveiled that HIV targets the immune cells, specifically CD4 T-cells. HIV evolves in the human body with a rate unseen in other retroviruses, exemplified from R5 (M tropic) to X4 (T tropic) virus. Under cART (Highly Effective Anti-Retroviral Treatment, or Combined Anti-Retroviral Therapy), there are founder virus and quasispecies [45,46]. Moreover, ample molecular studies have revealed that HIV 5′LTR governs the gRNA expression by the viral enhancer and promoter. Herein, we refresh our knowledge of HIV 5′LTR, to delineate the arsenal of the virus, and armamentaria of the host in control of HIV gRNA expression towards a cure. 

### 3.1. 5′LTR

HIV 5′LTR contains both enhancer and promoter elements [3,5,6,7,47,48,49,50,51,52,53,54,55,56,57,58,59,60,61,62,63,64,65,66,67,68,69]. Upon receiving cell signals and within a host DNA niche, the provirus can transcript its gRNA. If there is no cellular signal and there is no niche, there is no gRNA expression, or at the most, an aborted gRNA expression. Therefore, not every provirus is equal, and the host genome’s immunity dictates the outcome. 

*HIV enhancer*. More than 30 years ago, the NFkappaB core element was identified as the HIV enhancer [6]. Continuing studies have identified other core enhancer elements, such as PMA, the most common and potent phorbol ester, inducible GGGACTTTCC core enhancer element [48,49,50,51], and the core enhancer element CCAAT/(C/EBP) all in HIV LTR [60,61,62,63,64,65,66,67,68,69,70,71,72,73,74,75]. These studies unveil a trajectory that shows how HIV replicates in CD4 T-cells, in monocytes/microphages, and in central nervous system cells [6,58,59,60,61,62,63,64,65,66,67,68,69,70,71,72,73,74]. The HIV enhancer, located in the HIV LTR, has played an indispensable role in determining the viral tropism. In other words, the HIV enhancer has dictated a range of cell types to host the viral RNA expression, therefore the viral tropism and the subtype formation [29,30,45,46,47,48,60,61,62,63,64,65,66,67,68,69,70,71,72,73,74,75]. 

Based on studies of cellular enhancers, the HIV enhancers arranged in the 5′LTR should be considered as the super enhancer and the intragenic enhancer [55,76]. Super enhancer is a region comprising multiple enhancers that is collectively bound by an array of transcription factor proteins to drive transcription of genes involved in cell identity. Note that enhancer functions at both directions of 5′ and 3′ in cis, which is different from the promoters.

The studies of HIV/AIDS have shown that HIV enhancer differs from cellular enhancer in its intragenic way, meaning HIV super enhancer drives only a gRNA expression from its 5′UTR and ends in the 3′UTR, for both the gRNA and its spliced mRNA. In contrast to the host cellular enhancers, which activate their target gene expressions by topological-spatial contact, usually, several kilo- to megabases away from the target genes to be activated. 

Finally, HIV enhancer confers a rate of evolution to a given HIV. This highly conserved region establishes a trajectory of a given HIV species on a successful infection of the host cells, starting at the attachment, entry, reverse transcription, integration, transcription, and, thereby, encapsidation, packaging, virion releasing, and a repeat of this lifecycle. 

In HIV infection, the dual LTR is capable of breaching the genome’s immunity to replicate HIV within intestinal macrophages and CD4 T-cells, from peripheral monocytes and CD4 T-cells to the neural microglial cells, astrocytes, and CD4 T-cells. The HIV enhancer may be the most effective weapon in the viral arsenal. In fact, the HIV super enhancer provides a 3D-platform to recruit not only cellular proteins/factors, but also viral proteins/factors, cooperating and overlapping with those on the HIV promoter to drive the viral replication in different tissues and cells. This prevalent tactic of HIV has breached the immunity of the host genome and allowed a parasite to live in our DNA, and its replication occurs at the expense of our lives. 

*HIV promoter*. The HIV promoter contains the basic sequences for RNA transcription. Similar to its cellular counterpart, the HIV promoter has basal level transcriptional activity. The proteins binding in this region interact with those in the super enhancer to transcribe the gRNA. The transcription of gRNA, however, needs the enhancer function [3,6,52,57,58,59,60,61,62,63,64,65,66,67,68,69,72,74,75]. 

As stated previously, the HIV super enhancer competes with the host genome’s immunity in RNA transcription. Reports have shown that a R5 virus has different arrangements of the main and other core enhancer elements than a X4 virus in the 5′LTR super enhancer region [6,47,58,60,62,64,65,70,71,72,73,77,78]. Of note, the interaction of protein–nucleic acid complex between enhancer and promoter responds, and regulates the host cell signals. Enhancer binding proteins usually accelerate or magnify the cellular signal, and drive the promoter to generate a transcription niche in the DNA genome via binding and interacting with transcription factor complexes at both enhancer-promoter sites to transcribe the gRNA. After this activation point, the rest of viral replication is like a ball falling on ground. The metastable equilibrium between host and provirus is broken. The ball has fallen on ground rather than remaining in a biologically metastable state. This is similar to a system that returns to equilibrium after small (but not large) displacements in the physical chemistry, and may be represented by a ball resting in a small hole on top a hill. If the ball is only slightly pushed, it will settle back into the hole. But a stronger push may start the ball rolling down the hill. 

### 3.2. 3′LTR

Although in symmetry with and being a duplicate of 5′LTR, the 3′LTR guides the effective gRNA expression by its polyA signal, or responds to cellular stimuli to stop the gRNA transcription [79,80,81], and, finally, functions as the 5′LTR when 5′LTR is defective due to the integration [82]. 

Note that the 3′LTR polyA signal, which shows a metastable feature in 3′UTR, is reported to form a transcription-dependent gene loop with the 5’LTR occluded polyA signal. Activation of 5’LTR polyA signal or inactivation of 3’LTR polyA signal abolishes the gene loop formation [83].

Recent studies reveal that the 3′LTR serves as the 5′LTR in the DNA negative strand, and transcribes an antisense ncRNA named ASP (the ncRNA codes the antisense protein ASP) [84,85]. The ASP regulates the HIV latency by RNA:DNA base pairing to 5′LTR with Watson–Crick and Hoogsteen specificity. After the pairing, the provirus is back in business: latency. This epigenetic silencing differs from RNAi in the molecular mechanism: the ASP associates with polycomb repressor complex and promotes nucleosome assembly for the transcriptional-epigenetic silencing.

### 3.3. Impact of LTR and UTR in Viral Tropism

Both LTR and UTR play important roles in determining HIV infection of CD4 T cells or macrophages. The HIV LTR, via the enhancer that binds the host cellular transcription factors, transcribes the viral RNA. The 3′UTR is responsible for the viral RNA stability and then the viral protein expression. Together, they drive an active viral replication and the production of CXCR4 (X4) or CCR5 (R5) viruses. 

The interaction of host cellular transcription factors with LTR, i.e., HIV enhancer, determines the viral mRNA and gRNA expression. Despite the fact that the repeats of the enhancer elements vary in different HIV strains, such as R5 or X4, or a dual-tropic HIV [66,69,70,71,72,73], the NFkB core elements are considered as the main enhancer, and other enhancers add to the tissue and cell specific expression of gRNA [6,58,59,60,61,62,63,64,65,66,67,68,69,70,71,72,73,74,86]. 

Similar to its cellular counterpart, HIV enhancer is a driver for the cell and tissue specific expressions of HIV RNA, and, therefore, controls the HIV tropism [29,30,45,46,47,48,55,76]. This underlines a molecular mechanism that adds onto the viral tropism, which used to refer an expression of cell surface co-receptors for the entry of R5 or X4 viruses. 

The interaction of host cellular proteins with 3′UTR allows a given virus to express viral proteins after entry. Cellular miRNAs and host proteins in a particular cell host have been shown to function on the viral 3′UTR [35,36,37,38,39,40,42,43,44]. The immune pressure exerted on HIV 3′UTR, in the form of miRNAs and cellular factors, determines whether the HIV protein is translated or not and, therefore, determines an HIV protein function, and, hence, the viral tropism—a host cell that HIV can make an active replication within [39,40,42,43,44].

Note here is that the interaction of viral RNA-host protein underpins the molecular mechanism of viral tropism, at the step targeting the function of viral mRNA on the viral protein translation. Hence, the interaction of RNA-protein at the viral 3′UTR determines the functions of viral mRNA and viral protein, determines the viral tropism, and embodies that a virus can make an active replication within a cell or not. This clearly differs from a viral entry via a cell surface co-receptor and differs in an intrinsic mechanism that regulates the viral protein expression. 

Lastly, the expression and function of ncRNA have been known to be the cell and tissue specific [87,88,89]. Taken together, studies have shown that the HIV 3′UTR, its ncRNA, and the cellular ncRNA not only play roles in formation the so-called T tropic, M tropic, R5, X4, and the founder virus or quasispecies of HIV, but also on the host immunity against these HIV strains [29,30,45,46]. 

## 4. Armamentaria of the Human Host 

The study of how HIV RNA is expressed in the nucleosome in vitro was conducted in the same period as that defining the HIV enhancers and promoters: in the DNA era [90,91,92], during the NIH launched Human Genome Project (HGP) that began in 1990 and completed in 2003. The technology and knowledge on nucleosome, i.e., the study of chromatin, however, has developed rapidly in this era of epigenetics (the transcriptome era) [10,93,94,95,96,97,98], symbolized by the Encyclopedia of DNA Elements (ENCODE) project started in 2003 and concluded in 2012. ENCODE analyzes and identifies the functional elements in the human genome sequence after the HGP. We refresh the knowledge of HIV/AIDS research to harness an unprecedented model on the loss of function in humans to define not only the cause but also the effect—a cure of HIV/AIDS. We need to pay back this unparalleled model in the study of the human immune diseases, and we urgently need to harness what this model has provided and further the discovery of a cure of HIV/AIDS and other immune maladies. 

### 4.1. Epigenetic Immunity

Built on the HIV/AIDS model, we have proposed that the host DNA has an evolutionally developed immunity against pathogenic nucleic acid invasion. We have named this immunity the epigenetic immunity, grounded on our and others studies that p21 (p21^Cip1Waf1Sdi1^) restricts the HIV infection of the human primitive hematopoietic cells, macrophages, and the CD4 T-cells of elite controllers [9,99,100,101,102]. Our study, showing that p21 prevented HIV infection in human primitive hematopoietic cells, establishes that human cellular mechanisms have an essential role to play in inhibition of HIV replication [9]. 

The epigenetic research on the pathogenesis of viral infections has, in fact, proceeded with the study of HIV/AIDS. The seminal studies in this area include, but are not limit to, the following: (**1**) the studies on the human cytomegalovirus, Epstein–Barr Virus, HIV, and foreign DNA insertion [97,103,104,105]; (**2**) the epigenetics of infectious diseases [106,107,108]; (**3**) the epigenetic regulation of immune cell memory [109,110]; (**4**) the epigenetic molecular study of CD4 T-cells [111,112,113,114,115,116,117,118,119,120]; (**5**) super enhancers [117,121,122]; (**6**) the function of non-coding RNAs (ncRNAs) [123,124]; and (**7**) the epigenetic repression or imprinting (DNA methylation) on the human endogenous retroviruses (HERVs) [98,125,126]. 

Building on the contributions above, we conclude that the epigenetic immunity consists of three elements: DNA methylation, histone modification, and ncRNA function (Figure 2). We believe that this immunity exists in every cell that has a genomic DNA, but not the cells without the genomic DNA, such as the mature red blood cells [9,10,127]. Similar to the known innate, adaptive, or trained immunity, the immunogens such as the pathogenic RNA and DNA can elicit the host epigenetic immunity. We have proposed a cVaccine, not only as a tool to study the epigenetic immunity, but also the immunogen to elicit the host epigenetic immunity. 

### 4.2. Chromatin Vaccine (cVaccine)

cVaccine is a functional gene transcription unit with enhancer, in nucleosome format that resists nuclease degradation while mediating epigenetic silencing of viral RNA by ncRNA function and the enhancer decommissioning process [128,129,130,131].

It is well recognized that HIV infects naïve CD4 T-cells. CD4 T-cells have the stem cell property of self-renewal and differentiating into effector cells. The naïve CD4 T-cells analogize embryonic stem cell asymmetric division upon an antigen stimulation, and differentiate into memory and immune effector cells. Most importantly, these cells are distributed into all body tissues and embody antigen specific lineage of CD4 cytotoxic T-cells (CTL), CD4 regulatory T-cells (Treg), CD4 follicular helper T-cells (Tfh), etc., to directly attack pathogen, regulate CD8 T-cell, B cell, and other immune cell functions. 

The enhancer decommissioning occurs routinely in stem cells, specifically in lineage differentiation into tissue specific effector cells by regulating the exact same genomic DNA. cVaccine is an immunogen (antigen), and functions as a primer of naïve CD4 T-cell enhancer, by triggering the signaling pathways of previously identified Toll Like Receptor (TLR), interferon and NF*k*B, to load the TF (transcription factor)/RNAPII to the CD4 T-cell enhancer. Such a primed cell enhancer allows CD4 T-cells to rapidly differentiate into effector cells upon encounter with the HIV. The antigen specific differentiation of immune cells by primed enhancers or promoters is dubbed the poised ones, and is well studied in the immunology [110,111,112,113,114,115,116,117,118,121,122].

The enhancer decommissioning between HIV and CD4 T-cell is the ectopic or cognate binding the ration (limited supply) of TF/RNAPII either to a viral super enhancer or a cell linage enhancer. Note that the integration of intrinsic signals and TF/RNAPII binding to the CD4 gene (Cd4) enhancer allows the CD4 lineage enhancer to become a super enhancer, and differentiates into the antigen specific immune cells [118,121,122]. Therefore, cVaccine in an enhancer decommissioning process elicits the cell genomic determined fate for CD4 T-cells by epigenetic regulations, ushers CD4 T-cell lineage development by its lineage enhancer transiting into the supper enhancer, which superiorly allows the naïve CD4 T-cells to differentiate into the immune effector cells. 

In other words, cVaccine serves as a primer to the CD4 T-cell lineage enhancer, and to prime the CD4 T-cell enhancer into an active and poised state, ready to differentiate to antigen specific CD4 T-effector cells upon reencounter with the HIV. The vaccinated CD4 T-cells counteract the HIV proviral enhancer (usually one provirus per a cell) not only by competitively loaded TF/RNAPII on their enhancers, but also by the eRNA (enhancer RNA) function. Both converge to silence the HIV RNA expression while allow CD4 T-cells to differentiate into anti-HIV effector cells to establish a long lasting anti-HIV immunity. This embodies the genomic immunity—epigenetic immunity. We consider this is a molecular mechanism of the low viral load post cART, in the HIV controllers and in the HIV latent infection. 

Compared to the current mRNA vaccine, the cVaccines stops the viral RNA expression rather than expressing a viral protein. cVaccine aims to stop the viral RNA expression and, therefore, stops all the viral protein expressions, cuts down the viral mutagenesis, and ceases toxicity of viral RNA and proteins from damaging our body. Viral RNA and proteins are the culprits causing DNA mutation, immune inflammation and autoimmunity. We term these deviations from anti-viral immunity as the hypersensitivity V herein [132].

Since the cVaccine resists the nuclease degradations, it could be administered by the routes of viral infections, from entry and all the way to the target cells per se, aiming to elicit the IgA, IgM, and IgG, as well as innate, adaptive, trained, and epigenetic immunity in a systemic manner. This results from the study of HIV/AIDS, which is a naturally occurred model of loss-of-function in human immunity. Such a model has empowered the research on human immunodeficiency without initiating any ethical disputes, at a scale from bench, in silico, to bedside, from molecular to systemic mechanis ms that unveil the human immune response in etiology, pathogenesis, and up to a cure.

## 5. Conclusions

We have stated four of the achievements that multidisciplinary investigators have contributed from the HIV/AIDS and epigenetics research fields: 1. HIV super enhancer is essential for the HIV RNA expression. 2. Host genome has an immune system protecting our DNA against the viral infection, embodying the epigenetic immunity of our DNA. 3. The epigenetic regulation has forced the appearance of different HIV tropisms via an LTR-UTR circuit governed by LTR—the HIV super intragenic enhancer. 4. Host immunity and cART have kept the HIV in a metastable stage—the provirus. Further studies on epigenetic silencing can enforce HIV to become a new endogenous retrovirus that is unable to cause disease for generations. 

## Figures and Tables

**Figure 1 viruses-14-01084-f001:**
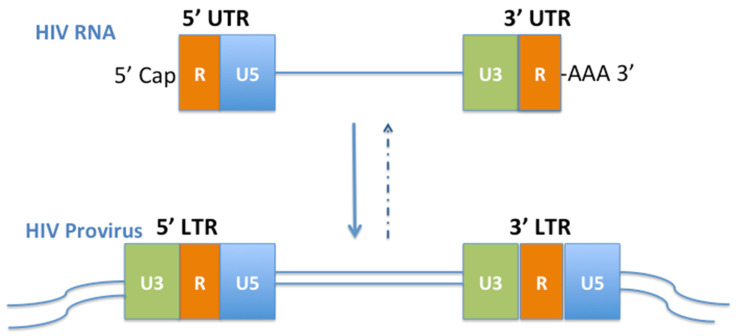
HIV UTR and LTR. U: unique element. R: repeat element. Provirus is a metastable stage in the retroviral lifecycle. A breach of the metastable equilibrium depends on the host cell signals not the virus. Based on the knowledge of X-chromosome inactivation and the stem cell features of CD4 T-cells, the host epigenetic silencing and cART can force the provirus into a stable state—a permanent silencing, similar to the ancient human endogenous retroviruses (HERVs) resided in our DNA.

**Figure 2 viruses-14-01084-f002:**
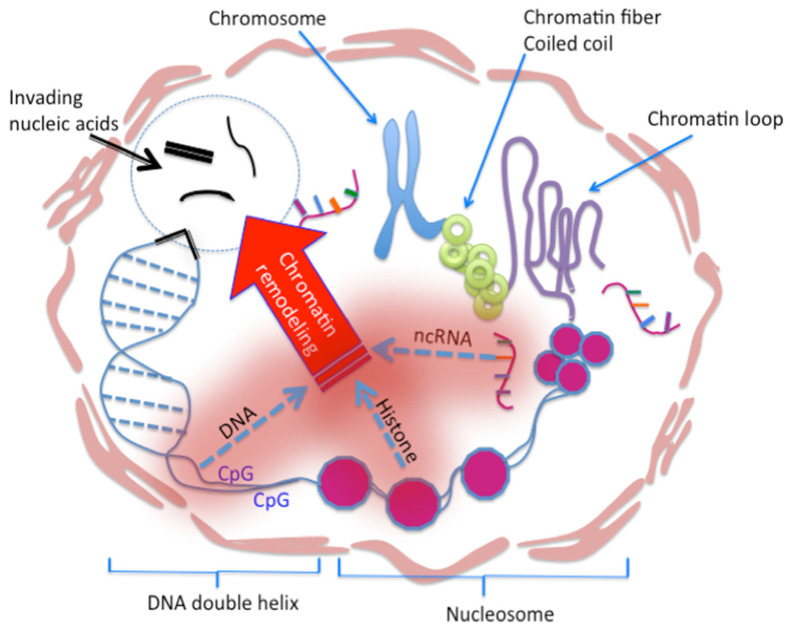
Epigenetic Immunity—a genetic immunity. A cell defends its genome against a foreign agent invasion by epigenetic immune regulations [9,10,127]. In HIV infection, the host cell protects its DNA from the viral nucleic acid attack by the epigenetic immunity, consisting of DNA methylation, histone acetylation/methylation, and non-coding RNA (ncRNA) activity. DNA methylation affects the function of double stranded DNA, histone modification affects the function of nucleosome, and ncRNA affects the function of chromatin. Each works independently but synergistically with the other two, guarding the topological structure and function of the DNA, and acting as a writer, reader, eraser, adaptor, modifier, organizer, or programmer [10,118,127]. The epigenetic factors reprogram genetic immune responses embodying gene activation, silencing, epigenetic memory, and chromatin remodeling in responding to environmental stimuli, specifically pathogenic foreign nucleic acids such as HIV.

## Data Availability

Data sharing is not applicable to this article as no datasets were generated or analyzed during the current study.

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
