# Peer review of "HIV UTR, LTR, and Epigenetic Immunity"

_viruses, 2022, doi:10.3390/v14051084_

Round 1

Reviewer 1 Report

Overall the manuscript summarizes a range of regulatory mechanisms acting on key HIV genome elements (i.e. HIV 5’- and 3’- UTRs and LTRs) and the resulting influences on the viral lifecycle. The authors primarily focus on how HIV UTRs and LTRs, present in HIV genomic RNA and integrated proviral DNA, respectively, impact viral transcription and translation, and as a result, tropism. In summarizing the impact of these elements on the viral life cycle, the authors focus on an alternative approach to suppressing HIV infection and latency, a chromatin vaccine (cVaccine). This cVaccine is presented as a more effective approach to suppressing HIV infection as it does not express a viral protein, but instead sequesters transcriptional machinery away from viral enhancers and primes CD4 T-cell lineage enhancer, thereby driving a robust immune response.

While the authors effectively highlight relevant epigenetic events occurring at the aforementioned viral elements and present a novel approach to suppressing HIV using a cVaccine, the manuscript would benefit from some edits before final publication.

Major comments:

  1. There are several instances where the authors meaning is not entirely clear and may need to be reworked for clarity. Some examples include line 174-176, 216-218, 250-252, 270-271. While some of these listed examples are made clear while reading further along, there might be some confusion for readers new to the field.
  2. Overall many parts of the manuscript get quite repetitive and could be condensed for reader clarity. A primary example of this is how often the authors mention that mechanisms acting at the UTRs (and LTRs) are important determinants of cellular tropism. This occurs at line 113-115, 130-132, 135-137, 138-140, 143-146, 201-203, 218-220. It may be easier on readers to condense these impacts on tropism into a small section.
  3. There are many instances where the exact mechanism being outlined is not clear. For example, in line 96-97 it is mentioned that eIF3f targeting to HIV 3’UTR restricts HIV mRNA expression. It may be worth mentioning that binding here blocks 3’ end processing. Similarly, it is mentioned that YTHDF binding to m6A modifications on HIV RNA leads to enhanced HIV protein expression and viral replication. It may be worth expanding on this and explaining how YTHDF binding enhances HIV protein expression - YTHDF2 has recently been shown to increase HIV mRNA stability while YTHDC1 reader recruitment helps regulate HIV alternative splicing (doi:10.1101/gad.348508.121).

Minor comments:

  1. Often acronyms are introduced without being defined (e.g. TAR- line 52; caRT-line 168) or are defined and defined again later (e.g. cVaccine defined in the abstract and line 27, and the definition repeated at line 301).
  2. Article usage in the manuscript is a bit off throughout the manuscript. For example, in line 28, “the” in “...against the HIV infection” is unnecessary.
  3. In Figure 1 it would be worth defining “R” and “U” labeled elements in the figure as “repeats” and “unique” elements respectively.
  4. Line 288: should read “...but are not limited to....”
  5. Line 349: should read “....while allowing CD4 T-cells to...”
  6. Line 359: “We term these deviations from anti-viral immunity as the hypersensitivity V herein.” It is not clear why this line was needed as hypersensitivity V is never used again in the manuscript.
  7. Line 290: “....2) the epigenetics of infectious disease...” It is unclear if this is referring to the epigenetic status of viral genomes (or the host genome) or if this refers to epigenetic mechanism acting against the virus.

Reviewer 2 Report

Review Report Form

Journal: Viruses (ISSN 1999-4915)

Manuscript ID: viruses-1670655

Type: Review

Title: HIV UTR, LTR and Epigenetic Immunity

Authors: Jielin Zhang * , Clyde Crumpacker *

Section: General Virology

Special Issue: Regulatory Mechanisms of Viral UTRs

The epigenetic immunity and HIV regulation pathways is a hot topic of the current scientific research. The HIV cure is a main objective in the fight against HIV/AIDS pandemic and the authors have published few articles on this subject in the previous years.

Comments

  1. The type of article is a review. The selection methodology of the cited articles should be described.
  2. A large amount of literature on HIV epigenetic is recently published, including experimental research and clinical trials. In my opinion, some of them are references studies and should be considered in this review article:
  • Shrivastava, S., Ray, R.M., Holguin, L. et al.Exosome-mediated stable epigenetic repression of HIV-1. Nat Commun 12, 5541 (2021). https://doi.org/10.1038/s41467-021-25839-2
  • Gerlinde Vansant, Heng-Chang Chen, Eduard Zorita, Katerina Trejbalová, Dalibor Miklík, Guillaume Filion, Zeger Debyser, The chromatin landscape at the HIV-1 provirus integration site determines viral expression, Nucleic Acids Research, Volume 48, Issue 14, 20 August 2020, Pages 7801–7817, https://doi.org/10.1093/nar/gkaa536
  • Einkauf KB, Yu XG, Lichterfeld M et al. Parallel analysis of transcription, integration and sequence of single HIV-1 proviruses.Cell 185: P266-282.e15, 2022 (open access). http://doi.org/10.1016/j.cell.2021.12.011.
  • Nguyen K, Dobrowolski C, Shukla M, Cho W-K, Luttge B, Karn J (2021) Inhibition of the H3K27 demethylase UTX enhances the epigenetic silencing of HIV proviruses and induces HIV-1 DNA hypermethylation but fails to permanently block HIV reactivation. PLoS Pathog 17(10): e1010014. https://doi.org/10.1371/journal.ppat.1010014
  • Oriol-Tordera B, Esteve-Codina A, Berdasco M, Rosás-Umbert M, Gonçalves E, Duran-Castells C, Català-Moll F, Llano A, Cedeño S, Puertas MC, Tolstrup M, Søgaard OS, Clotet B, Martínez-Picado J, Hanke T, Combadiere B, Paredes R, Hartigan-O'Connor D, Esteller M, Meulbroek M, Calle ML, Sanchez-Pla A, Moltó J, Mothe B, Brander C, Ruiz-Riol M. Epigenetic landscape in the kick-and-kill therapeutic vaccine BCN02 clinical trial is associated with antiretroviral treatment interruption (ATI) outcome. EBioMedicine. 2022 Mar 21;78:103956. doi: 10.1016/j.ebiom.2022.103956. Epub ahead of print. PMID: 35325780; PMCID: PMC8938861.
  1. Self-citation is found in 6 references. It could be excessive.
  2. A synthetic table on epigenetic regulation of un-translated terminal region (UTR) and the viral DNA in the form of long terminal repeat (LTR) should be added.
  3. The idea of chromatin HIV vaccine is not a new hypothesis considered by the authors. It should be presented as potential opportunities and limits.
  4. The figure 1 has no legend.
  5. The row 363: “This results from the studies of HIV/AIDS model.” Which model? It should be cited.
  6. Abbreviation list could be useful.

Reviewer 3 Report

the introduction should be more extensive on the subject related to epigenetic immunity. You miss some papers related to  Chromatin and epigentic immunity in HIV. 

Reviewer 4 Report

This review manuscript by Zhang J and Crumpacker explores HIV untranslated terminal region, long terminal repeats, and host epigenetics regulations. This manuscript improves a lots from the previous version. The authors have also updated the references. They have addressed my criticism. 

Round 2

Reviewer 2 Report

No

This manuscript is a resubmission of an earlier submission. The following is a list of the peer review reports and author responses from that submission.

Round 1

Reviewer 1 Report

Please update the references and make statements that are scientifically sound.

Reviewer 2 Report

This review manuscript by Zhang J and Crumpacker explores HIV untranslated terminal region, long terminal repeats, and host epigenetics regulations. I found this review very interesting however there are some drawbacks

  1. Please extend the abstract by redrafting a new abstract.
  2. The introduction part is not well described. It should serve the purpose of leading the reader from a general subject area to a particular field of research. Please consider rewriting the introduction section.
  3. This review has a very strange way of citing other research work. It is kind of unusual the whole introduction has no reference at all. It is not ideal to write a whole paragraph and insert the reference at the end. I would suggest the author please cite in such a way that the readers can easily understand the original source of information or the statement.
  4. Line no.21-22 “We will review the findings of many investigators to assemble this knowledge in a holistic manner, to generate effective immunogens, and to elicit the immunity of DNA against the HIV infection”. I am confused about this statement. What effective immunogens have been generated?
  5. Line no 138-140, Under “cART (Highly Effective Anti-Retroviral Treatment, or Combined Anti-Retroviral Therapy), there are founder virus and quasispecies” please cite this statement. Moreover, I would request the authors to kindly elaborate on this.
  6. It will be interesting if authors could extend more about cVaccine comparing in-depth details with mRNA vaccine. Please add more on this.

Reviewer 3 Report

Zhang and Crumpacker in this review proposes what is an interesting concept, that epigenetic regulation of HIV-1  proviruses may act as an "innate" intracellular restriction of proviral expression and replication. Despite the potential provocative hypothesis, the review suffers from being overly superficial. For example, they do not ever really define the concept of epigenetic immunity or highlight potential mechanisms that could be driving this. Similarly, a chromatin vaccine is proposed but exactly what that means and how that would be achieved is not discuss. Key concepts of HIV reverse transcription, transcriptional control, current understanding of epigenetic regulation of HIV, and restriction factors that do restrict HIV replication are not fully explored and often not even mentioned. The authors do discuss tropism and transcription at some length, but it was not clear if the authors were really suggesting transcription and entry were coupled or if they were proposing cell specific factors can drive the evolution of these transcriptional elements which is independent of actual tropism. Finally, the paper lacks appropriate and current references. There are few recent references which is surprising because understanding epigenetic regulation of HIV and its impact on persistence is a very active topic in HIV cure research. General statements are made that lack references and then there are sections that include 10 or so references but it is unclear as to what those references are supporting. In summary, this review reads like a set of bullet points of vague concepts and speculated functions without context or support from the literature while ignoring more recent mechanistic insights into HIV transcription and translation.